# Evidence for Oxidative Pathways in the Pathogenesis of PD: Are Antioxidants Candidate Drugs to Ameliorate Disease Progression?

**DOI:** 10.3390/ijms23136923

**Published:** 2022-06-22

**Authors:** Alexander Leathem, Tamara Ortiz-Cerda, Joanne M. Dennis, Paul K. Witting

**Affiliations:** 1Charles Perkins Centre, School of Medical Sciences, Faculty of Medicine and Health, The University of Sydney, Sydney, NSW 2006, Australia; zander.leathem@gmail.com (A.L.); tamara.ortizcerda@sydney.edu.au (T.O.-C.); jodennis121@gmail.com (J.M.D.); 2Department of Normal and Pathological Cytology and Histology, Faculty of Medicine, University of Seville, 41009 Seville, Spain

**Keywords:** Parkinson’s disease, neurodegeneration, oxidative stress, cyclic nitroxide, mitochondrial dysfunction, reactive oxygen species, antioxidants

## Abstract

Parkinson’s disease (PD) is a progressive neurodegenerative disorder that arises due to a complex and variable interplay between elements including age, genetic, and environmental risk factors that manifest as the loss of dopaminergic neurons. Contemporary treatments for PD do not prevent or reverse the extent of neurodegeneration that is characteristic of this disorder and accordingly, there is a strong need to develop new approaches which address the underlying disease process and provide benefit to patients with this debilitating disorder. Mitochondrial dysfunction, oxidative damage, and inflammation have been implicated as pathophysiological mechanisms underlying the selective loss of dopaminergic neurons seen in PD. However, results of studies aiming to inhibit these pathways have shown variable success, and outcomes from large-scale clinical trials are not available or report varying success for the interventions studied. Overall, the available data suggest that further development and testing of novel therapies are required to identify new potential therapies for combating PD. Herein, this review reports on the most recent development of antioxidant and anti-inflammatory approaches that have shown positive benefit in cell and animal models of disease with a focus on supplementation with natural product therapies and selected synthetic drugs.

## 1. Introduction

Parkinson’s disease (PD) is a progressive long-term neurodegenerative disorder in which the primary pathological feature is selective loss of dopaminergic neurons in the substantia nigra pars compacta and the presence of α-synuclein containing Lewy bodies [1]. PD arises due to a complex interplay between age, genetic, and environmental factors with the mechanisms underlying the loss of dopaminergic neurons thought to include oxidative stress, mitochondrial dysfunction, and inflammation, as well as genetic mutations and abnormal handling of misfolded proteins by the ubiquitin-proteasome and autophagy-lysosome systems, all leading to a characteristic impairment of physical motor function [2]. Treatment of PD is primarily centered on the use of levodopa and deep brain stimulation [3]; however, these therapeutic approaches produce only symptomatic relief and there is a growing need to develop new approaches to address the underlying pathogenesis of the disease, especially as PD will pose an increasingly significant health concern in Western countries as life expectancy continues to rise [3]. Oxidative stress is a prominent pathophysiological mechanism implicated in PD and accordingly, there is significant research into the development of antioxidants as potential disease-modifying therapies. Herein, we review the role of oxidative stress in PD pathogenesis and the efficacy of both natural endogenous and synthetic antioxidants (refer to summary Figure 1).

## 2. Evidence Supporting Enhanced Oxidative Stress in PD

Oxidative stress is proposed to contribute significantly to the dopaminergic neuron loss seen in both idiopathic and genetic PD. This is supported by the presence of oxidized lipids, proteins, and DNA as well as decreased concentrations of reduced glutathione (an antioxidant) in the post-mortem substantia nigra tissue of PD patients [4,5,6]. Excessive production of reactive oxygen species (ROS) is detrimental to cell survival and normal function as these ROS oxidize cellular macromolecules, mutate mitochondrial DNA, and induce caspase activation and mitochondrial-mediated apoptosis via the release of cytochrome c and other pro-apoptotic proteins [7]. Similarly, ROS participate in the sphingomyelin- or ceramide-dependent signaling pathway that results in the translocation of the pro-inflammatory transcription factor nuclear factor kappa B (NF κB) from the cytoplasm to the nucleus where it transcriptionally activates pathways linked to the induction of apoptosis [8]. Increased oxidative stress disrupts the ubiquitin-proteasomal system leading to an accumulation of damaged and misfolded proteins, a pathophysiological mechanism that is also observed in PD [9].

### 2.1. Mitochondrial Dysfunction

In normal physiological conditions, a balance is maintained between ROS production and antioxidative cell defense mechanisms that prevent oxidative stress. However, dopaminergic neurons are particularly sensitive to oxidative stress due to factors such as significant reliance on aerobic respiration for ATP production, high concentrations of intracellular iron, which participates in the oxygen radical producing Fenton reaction, and ROS-producing enzymes involved in normal dopamine metabolism, such as tyrosine hydroxylase (TH) and monoamine oxidase [10].

Neurons are heavily dependent on aerobic respiration for ATP and consequently, mitochondrial dysfunction can lead to a substantial loss in energy reserves with a parallel increase in ROS production that exceeds the capacity of antioxidant defense mechanisms and promotes a loss in cell viability [10]. Herbicide chemicals such as paraquat, rotenone, and 1-methyl-4-phenylpyridinium (MPP+) are associated with the environmental aspects of PD etiology [11]. These induce permanent PD-like symptoms by inhibiting complex I in the respiratory chain, resulting in an increased electron leakage to oxygen and subsequent production of ROS which increases intracellular oxidative stress [11]. Similarly, although α-synuclein remains mainly cytosolic, it can interact with mitochondrial membranes and inhibit electron transport via complex I [12].

In addition to environmental factors, mutations in mitochondrial protein-encoding genes have been identified in familial forms of PD. Mutations or deficiency in the *parkin* gene result in reduced mitochondrial membrane potential and decreased complex I activity, Phosphatase and tensin homolog (PTEN) Induced Kinase–1 (referred to as PINK–1) mutations increase free radical formation and induce mitochondrial dysfunction, and mutations that affect DJ–1, an atypical peroxidase that scavenges hydrogen peroxide, result in increased oxidative damage [13,14,15]. Additionally, reduced complex I activity and an increased proportion of dopaminergic neurons with respiratory chain defects have been observed in the post-mortem substantia nigra tissue of sporadic PD patients, supporting the implication of mitochondrial dysfunction in the pathogenesis of the disease [16,17].

### 2.2. Iron Dysregulation

Increased accumulation of iron in the putamen, globus pallidus, red nucleus, and substantia nigra is a feature of normal ageing [18]. However, this increase is significantly amplified in PD patients and may result from increased iron influx, dysregulation of intracellular homeostasis of iron metabolism, or reduced iron efflux [19]. Intracellular ferric iron is usually contained in mineral ferrihydrite form by a ferritin protein cage; however, the release of soluble ferrous iron from ferritin is a normal feature of ageing and furthermore, homogenates of diseased substantia nigra tissue have been observed with decreased levels of ferritin [20,21]. Similarly, amyloid precursor protein and soluble tau protein, which are involved in the ferroportin-ceruloplasmin export mechanism are decreased in the substantia nigra of PD patients [22,23].

Dysfunction in iron regulatory control mechanisms in PD can yield an increase in unbound ferrous iron, which exerts neurotoxicity by several mechanisms. For example, iron-mediated oxidation of dopamine produces dopamine quinone, which modifies a variety of cellular structures, and 6-hydroxydopamine (6–OHDA), a potent inhibitor of mitochondrial complexes I and IV, is commonly used in experimental models to induce parkinsonian-like properties [24]. Furthermore, ferrous iron reacts with hydrogen peroxide, a by-product of 6–OHDA metabolism and normal mitochondrial respiration, in the Fenton chemical reaction yielding the potent hydroxyl radical that increases intracellular oxidative stress and causes diffuse indiscriminate cellular damage via lipid peroxidation, membrane integrity disruption, and the formation of DNA adducts [25].

### 2.3. Dopamine Quinone

Dopamine is naturally stored in vesicles; however, excess cytosolic dopamine in an environment of oxidative stress is spontaneously and enzymatically oxidized to dopamine quinone which has several debilitating downstream effects [10]. Dopamine quinone covalently modifies cellular nucleophiles involved in cell survival such as low molecular weight sulfur-containing molecules (e.g., glutathione) and protein cysteinyl residues [10]. Furthermore, dysfunctional variants of proteins such as α-synuclein, parkin, protein deglycase-1 (DJ–1; also known as Parkinson’s disease protein 7), and ubiquitin C-terminal hydrolase (UCH–1) have been implicated in the pathophysiology of PD and these proteins are also modified by dopamine quinone [10]. Notably, dopamine quinone encourages the conversion of α-synuclein into its toxic oligomeric form which has several cytotoxic downstream effects and is involved in a positive feedback loop that propagates cellular damage [26].

Similarly, dopamine quinone participates in another damaging positive feedback loop as it permeabilizes vesicle membranes, increasing free cytosolic dopamine concentrations and subsequent conversion to dopamine quinone [27]. Dopamine quinone causes swelling in neural mitochondria and general mitochondrial dysfunction by modifying the subunits of complexes I and III, thereby increasing ROS production and the oxidative stress in the cellular environment [28]. Cyclization of dopamine quinone produces amino-chrome which polymerizes to form neuromelanin, producing superoxide and depleting NADPH in the redox process [10]. Neuromelanin acts as an inflammatory mediator that triggers microglia overactivation and an increased inflammatory state in the microenvironment, thereby aggravating neurodegeneration [29].

## 3. Antioxidants

### 3.1. Endogenous Low-Molecular Weight Molecules

#### 3.1.1. Coenzyme Q

Coenzyme Q_10_ (CoQ_10_) is a lipid-soluble antioxidant essential to mitochondrial electron transport chain (ETC) function and facilitates the transport of electrons from complex I and II to complex III [30]. As an antioxidant, CoQ_10_ directly reduces the initiation and propagation of lipid peroxidation by preventing lipid peroxyl radical production, as well as indirectly by regenerating other antioxidants such α-tocopherol (Vitamin E) and ascorbate (Vitamin C) [31,32]. CoQ_10_ is present in all intracellular membranes; however, as a component of the oxidative phosphorylation pathway, it is especially well placed in mitochondria, where electron leakage contributes to cellular superoxide and hydrogen peroxide production, to inhibit protein and DNA oxidation by both a general ROS scavenging mechanism as well as functioning as a chain-breaking antioxidant to prevent free radical propagation [32].

A significant deficiency of CoQ_10_ with functional ex vivo analysis has been shown in PD patients compared to age and gender-matched controls [33] as well as decreased total plasma reduced coenzyme Q_10_ (also referred to as ubiquinol_10_) and increased oxidized CoQ_10_, a marker of oxidative stress [34]. Similarly, post-mortem studies of brain homogenates found a significant reduction in total CoQ_10_ concentration in the cortex of the brain [35]. The redox state of CoQ_10_ affects its antioxidant and mitochondrial/ETC functions and hence could impact PD. Platelets from PD patients show decreased ubiquinol_10_:CoQ10 (reduced:oxidized) ratios [36], signifying increased oxidative stress, and low CoQ_10_ together with reduced complex I and II/III activities in mitochondria [37]. Furthermore, higher levels of oxidized CoQ_10_ and 8-hydroxy-2′-deoxyguanosine (8–OHdG) are found in the CSF of patients with PD compared to healthy controls [38]. Accordingly, a deficiency of CoQ_10_ in PD supports a potential therapeutic role for dietary supplemented CoQ_10_.

A neuroprotective effect of CoQ_10_ has been demonstrated in numerous in vitro PD studies. Thus, CoQ_10_ reduced oxidative stress and neuronal damage induced by paraquat in SHSY5Y cells [39] and has also been found to stabilize the mitochondrial membrane to reduce both total cellular and mitochondrial ROS generation and apoptosis induced by hydrogen peroxide in a neuronal cell model of PD [40]. The promising therapeutic potential of CoQ_10_ has been further established in in vivo studies with various PD models. Supplementing CoQ_10_ reduced substantia nigra dopaminergic neuron loss in an MPTP primate model [41] and similarly, in a paraquat mouse model, increasing CoQ_10_ elicited a substantial improvement in behavioral tests and decreased brain protein carbonyl content, a marker of oxidative stress [42]. More recently, CoQ_10_ cotreatment with a novel microRNA mimics reduced dopaminergic neuron loss and improved motor function in rats in a 6-OHDA induced PD model [43].

Clinical trials of CoQ_10_ have not replicated the promising results obtained in cell and animal models. Initial pilot studies and small clinical trials demonstrated that CoQ_10_ was safe and well-tolerated up to 1200 mg/d [44,45], and the minimal side effect profile of CoQ_10_ has also been substantiated in an unrelated study in Huntington disease as well as a more recent phase 3 RCT [46,47]. The Unified Parkinson’s Disease Rating Scale (UPDRS) is a clinical rating scale applied prior to 2008 in the assessment of PD severity. It was initially developed in the 1980s [48] and mainly considered aspects of non-motor experiences of daily living with PD (Part I). In 2008, criteria for the UPDRS were altered after further clinometric testing accounting for specific criteria for movement disorders yielded the revised UPDRS classifications of motor experiences of daily living (Part II), motor examination (Part III), and motor complications (Part IV) [49] with a renaming of the scoring system to the Movement Disorder Society UPDRS (referred to as the MDS-UPDRS). An early clinical trial with CoQ_10_ demonstrated a dose-dependent reduction in worsening of UPDRS; however, this study had a relatively small sample size of eighty enrolled patients, and its findings have not been reproduced in larger clinical trials [45]. The QE3 trial, the largest clinical trial of CoQ_10_ for PD to date, considered a participant population of six hundred and found that both 1200 and 2400 mg/d of CoQ_10_ demonstrated no clinical benefit, and in fact both treatment groups showed slight adverse trends in MDS-UPDRS scores relative to placebo [47]. The results of the QE3 trial have been substantiated by later meta-analyses, which concluded that current evidence indicates that CoQ_10_ does not slow functional decline, provide symptomatic benefit, or improved motor function [50,51].

Despite current evidence suggesting minimal symptomatic or disease-modifying effects of CoQ_10_, several promising novel therapeutic approaches have been identified, indicating that the potential clinical benefit of CoQ_10_ in PD has yet to be reached. For example, Park et al. demonstrated in a 6-OHDA rat model that continuous intrastriatal administration of low dose CoQ_10_ improved behavioral tests and decreased dopaminergic neuron loss, and notably, that despite a significantly lower dose these effects were more pronounced than those elicited by oral administration of CoQ_10_ [52]. The reduced form of CoQ_10_, which has been shown to have a better neuroprotective effect in animal models [53], was demonstrated to produce a statistically significant reduction in UPDRS scores in PD patients treated with a comparatively low dose of 300 mg/d as compared to the placebo [54]. Interestingly, the reduced form of CoQ_10_ tested did not elicit any clinical benefit in early PD patients who were not taking levodopa, suggesting that CoQ_10_ may be beneficial in specific PD patient populations, a nuance lost by studies with broader inclusion criteria [54]. Of note, a phase 2 clinical trial was registered in 2018 which will focus on employing an omics-based strategy to stratify PD patients into subgroups based on their genetic mitochondrial risk burden and their expected treatment response to CoQ_10_ [55]. The awaited results from this study could validate one of several promising novel directions for future research on the role of CoQ_10_ in the treatment of PD. 

#### 3.1.2. Urate

Urate is a prominent plasma antioxidant [56] that inhibits lipid peroxidation and damage to biological molecules by acting as a free radical scavenger that neutralizes singlet oxygen, superoxide radical anion, and peroxynitrite (ONOO-) [57,58]. There is substantial epidemiological evidence that high urate levels are associated with a reduced risk for developing PD as well as a slower rate of clinical decline, although interestingly, despite robust evidence in men, this trend is variably observed in women, reflective of the complex interplay of factors influencing PD pathogenesis [59,60,61].

Consistent with the observed epidemiological association in human PD, results from in vivo and in vitro studies support the potential therapeutic role of urate. In differentiated PC12 cells, urate inhibited 6-OHDA induced neurotoxicity and reduced oxidative stress, as assessed by biomarkers such as lactate dehydrogenase (LDH), malondialdehyde (MDA), and 8-OHdG [62]. Furthermore, urate pretreatment in dopaminergic cells (SH-SY5Y and MES23.5) attenuated 6-OHDA and hydrogen peroxide-induced cell death, and the mechanism underlying this neuroprotection was dependent on nuclear factor erythroid 2-related factor (Nrf2) activation and the upregulation of this transcription factor’s target antioxidant genes, γ-glutamate-cysteine ligase catalytic subunit (γ–GCLC) and heme oxygenase-1 (HO–1) [63]. Similarly, urate has been shown to induce the Nrf2 signaling pathway in astrocytes, eliciting neuroprotection against hydrogen peroxide through the elevation of glutathione [64]. Interestingly, urate and its precursor, inosine, were found to reduce the toxic effect of hydrogen peroxide as well as markers of free radical generation and oxidative damage in co-cultures of astrocytes and MES23.5 dopaminergic cells, but not in dopaminergic monocultures alone, suggesting that urate may provide multi-faceted neuroprotection in vivo [65,66]. 

Consistent with a neuroprotective mechanism identified in cell models, urate increased mRNA and protein expression of Nrf2 and its dependent antioxidant genes, γ-GCLC and HO–1, in mice treated with MPTP [67]. In the same study, urate was also found to increase concentrations of antioxidants such as SOD and glutathione in the SN as well as modulate neuroinflammation by inhibition of inflammatory markers such as IL-1β, IL-6, and TNF–α [67]. Furthermore, urate reduced dopaminergic neuron loss in rats treated with 6–OHDA, and this neuroprotective effect was attributed to regulation of the protein kinase B/glycogen synthase kinase 3 beta signaling pathway, as well as the established upregulation of endogenous antioxidants [68].

Clinical trials have been somewhat limited by the association of elevated urate and an increased risk of gout [69]. However, inosine, a urate precursor, has been found to be generally safe, well-tolerated, and effective in increasing serum and CSF urate levels as well as plasma antioxidant capacity [70,71,72]. The SURE-PD3 trial, a randomized, double-blind, placebo-controlled phase 3 trial of oral inosine treatment in two hundred and ninety-eight early PD patients closed early based on a pre-specified interim futility analysis [73]. This trial demonstrated no significant difference between inosine and the placebo in clinical progression or secondary efficacy outcomes such as dopamine transporter binding loss [73]. Although the current evidence does not support the use of inosine in the treatment of early PD, whether urate-elevating therapy will influence other stages in the disease progression remains to be seen. 

#### 3.1.3. Glutathione

The glutathione (GSH) system is the main endogenous antioxidant defense and protects cell viability and function by a mechanism of free radical scavenging, interaction with iron metabolism, and transition metal chelation and as a cofactor for other antioxidants, such as glutathione peroxidase [74]. GSH also has a role in preventing mitochondrial dysfunction by maintaining the redox status of proteins via glutaredoxins, as well as the chemical reduction of toxic dopamine quinone formed during dopamine auto-oxidation [75,76]. Consistent with the role of oxidative stress in PD pathogenesis, there is a decrease in reduced glutathione and an increase in the levels of oxidized glutathione in post-mortem analysis of substantia nigra tissue [77]. Furthermore, decreased erythrocyte GSH concentration is associated with more severe disease as judged by the original Unified Parkinson’s Disease Rating Scale (UPDRS) scores [78], and similarly, lower serum GSH is correlated with worse Montreal Cognitive Assessment (MoCA) scores and Hoehn-Yahr staging [79], suggesting these measures may have utility as biomarkers in PD.

N-acetylcysteine (NAC), a membrane-permeable cysteine precursor which functions as the rate-limiting substrate in GSH synthesis, is a potential therapeutic approach to target the GSH system, given that neurons cannot directly uptake GSH [80]. NAC has been found to ameliorate the decreased levels of GSH in the substantia nigra in a 6-OHDA hemi-parkinsonian rat model [81]. Similarly, intravenous infusion of NAC increases both brain GSH concentration and blood glutathione redox ratio in PD patients as well as healthy controls, although this effect is, as expected, more pronounced in the PD patient group [82]. However, the use of NAC in the treatment of PD is limited by a short half-life and poor oral bioavailability due to extensive first-pass metabolism [83]. Accordingly, an important direction of research is in the development of delivery systems with more favorable pharmacokinetics. For example, mitochondria-targeted NAC nanocarriers, which when formulated with NAC were shown, produce a particle size that enables blood-brain barrier BBB passage and reduces rotenone-mediated cell death and oxidative stress in SH-SY5Y cells [84]. 

Furthermore, scopoletin, a novel antioxidant moiety, restored redox balance and reduced oxidative damage and mitochondrial dysfunction in both MPP^+^ treated SH-SY5Y cells and a *Drosophila* genetic model of PD [85]. Interestingly, this mechanism of neuroprotection was mainly attributed to maintaining glutathione levels in its oxidizable form [83]. This suggests that the development of novel therapeutics which target the GSH system presents a promising approach to circumvent the pharmacological difficulties associated with administering GSH, or its precursor, NAC, while still harnessing the potent antioxidant capacity of this system. 

In clinical trials, a combination of intravenous infusions and oral doses of NAC was shown to significantly increase dopamine transporter binding in the caudate and putamen [86]. However, the unfavorable pharmacokinetics of oral NAC reduce the potential clinical utility of this approach. A prospective 4-week study found that although oral NAC increased peripheral antioxidants measures such as catalase and GSH/GSSG, this treatment failed to increase brain GSH or reduce markers of lipid peroxidation [87]. Furthermore, MDS-UPDRS scores increased in four of the five PD patients treated with oral NAC, a worsening of clinical parameters which the authors suggested may have occurred due to drug-drug interactions between NAC and other PD medications, such as amantadine [87]. A recent meta-analysis, which considered both intranasal and intravenous administration of GSH identified a statistically significant increase in glutathione peroxidase levels as well as an improvement in section III, the motor examination component of the under the revised MDS-UPDRS criteria [88]. Interestingly, the authors also performed a subgroup analysis [88] which found that 300 mg/d GSH was more effective than 600 mg/d in reducing MDS-UPDRS III scores. However, there is significant variation in both the dose and method of administration in clinical trials of GSH and further research is necessary to investigate both the optimal treatment strategy as well as to confirm any clinical benefit identified. 

### 3.2. Dietary Antioxidants

#### 3.2.1. Vitamin C

Vitamin C (ascorbate) is a water-soluble vitamin that functions as a co-factor in several biosynthesis pathways, including collagen, catecholamine, and neuropeptide production, as well as maintenance of redox homeostasis as a prominent antioxidant [89]. Ascorbate acts as a free-radical scavenger and antioxidant through the non-enzymatic reduction of ROS such as superoxide, hydroxyl, and peroxyl radicals [90]. Ascorbate also functions in the regeneration of Vitamin E, another antioxidant, by reducing the tocopheroxyl radical [90]. A recent study demonstrated a higher proportion of hypovitaminosis C in PD patients compared to healthy controls, and that this deficiency of ascorbate was associated with worse cognitive impairment and lower MoCA scores [91]. PD patients have significantly lower serum ascorbate [92], and furthermore, reduced lymphocyte ascorbate concentration is associated with increased PD severity, as assessed by Hoehn-Yahr staging [93]. 

Cell and animal studies of vitamin C support the therapeutic potential of this antioxidant in the treatment of PD. In a model of dUCH using *Drosophila*, where a homolog of the PD-related gene UCH-L1 was a knockdown, treatment with low dose ascorbate reduced dopaminergic neuron loss and motor dysfunction, although side effects on physiology occurred with high doses and long-term treatment [94]. Ascorbate also reduces ROS and inducible nitric oxide synthase (iNOS) in MPP^+^ treated astrocytes as well as upregulates endogenous antioxidants through a mechanism of NFκB activation in both astrocytes and MPTP treated mice [95]. Similarly, in another MPTP-induced PD mouse model, co-treatment with ascorbate and NXP031, a novel single-stranded DNA aptamer that increases the efficacy of ascorbate by binding and reducing its oxidation, ameliorated dopaminergic neuron loss and microglia induced neuroinflammation in the substantia nigra [96]. The use of novel adjunct agents is a promising direction of research to improve the efficacy of vitamin C.

A meta-analysis of twelve studies, which considered dietary intake of vitamin C, concluded that there was no significant reduction in risk of developing PD in the high vitamin C intake group as compared to the low intake group [97]. Furthermore, six of the studies included in this meta-analysis also found that supplemental vitamin intake did not have any effect on the development of PD [97]. However, ascorbate supplementation has also been shown to positively impact levodopa pharmacokinetics, improving both absorption and bioavailability, suggesting that the therapeutic role of ascorbate is not limited to reducing PD development [98]. Nevertheless, further investigation of ascorbate in clinical trials to assess its effect on disease progression and symptom burden is necessary to determine the potential therapeutic role of this antioxidant. 

#### 3.2.2. Vitamin E

Vitamin E (α-tocopherol) is the major lipophilic antioxidant in the brain. Chronic vitamin E deficiency in this organ coincides with lipid peroxidation, altered phospholipid composition, and energy metabolism together with impaired cognition in a recognized model of brain impairment [99]. The redox activity of α-tocopherol and its accumulation in bio-membranes enhances scavenging of chain-propagating free radicals to inhibit lipid peroxidation in phospholipid bilayers [100]. Other antioxidants such as vitamin C and CoQ_10_ act in a concerted manner to regenerate α-tocopherol to maintain cell/tissue redox balance thereby, recycling the active form of vitamin E in biological systems. Depletion of these co-antioxidants can limit the bioactivity of vitamin E and switch to a pro-oxidant action [100].

Severe vitamin E deficiency is rare, and it is currently unknown whether nutritional deficiency contributes to neurodegenerative disease. In addition to its endogenous antioxidant potential, vitamin E has been proposed as a therapeutic target in PD and other neurodegenerative diseases, as deficiency or malabsorption cause neurological syndromes such as ataxia and nerve damage. Low blood and CSF vitamin E concentrations are consistently observed in neurodegenerative disease [101] along with oxidative stress [102], and supplementation improves cognitive decline and reduces lipid peroxidation and amyloid deposition [103].

Vitamin E elicits beneficial effects in animal studies of PD. For example, supplementation slows dopaminergic neuron loss in a model of superoxide perturbation [104] and attenuates both behavioral and biochemical abnormalities after 6–OHDA challenge in rats [105]. Further, α–tocopherol protects neurons and alleviates locomotor symptoms in early PD disease models [106,107]. In a rat rotenone model, intervention with vitamin E significantly decreased lipid peroxidation and improved SOD, GSH, and dopamine levels, and these biochemical changes translated into reduced neurodegeneration and alleviation of motor deficits [108,109]. Chronic vitamin E also restored synaptic plasticity in PINK–1 K/O mice, a model of subclinical PD that shows decreased dopamine release in the absence of overt neuron loss [110]. This study suggested that neuroprotection resulted from vitamin E-induced improved mitochondrial bioenergetics like that observed with a water-soluble vitamin E analog (Trolox) and mitochondrial complex 1 and IV activity in PINK–1-deficient dopaminergic cells [111].

Modulation of neuroinflammation by vitamin E may also be protective in the pathogenesis of PD. A-Tocopherol influences inter/intracellular interactions and biochemical pathways involving membrane enzyme activity, signal transduction, and receptor function via the protection of bio-membranes against oxidation. It is likely that these cellular effects contribute to immune modulation in animal studies where vitamin E supplementation shows positive benefits on innate immunity. For example, vitamin E supplementation improves synapse formation and signaling in naive T-cells in vivo in aged animal studies [112] and enhances IL–2, immune cell activity, and immune responses [113]. Importantly, vitamin E also shows efficacy in improving T-cell function and immune responses in aged human studies [113]. These positive outcomes may be related to antioxidant activities that maintain bio-membranes and, therefore, effective signaling that is imperative for correct immune responses and/or via gate-keeping levels of lipid peroxidation products that act as inflammation mediators. Both animal and human studies show vitamin E supplementation reduces the proinflammatory mediator, prostaglandin E2 and other pro-inflammatory cytokines such as TNF–α and IL–6 [112]. Vitamin E also interacts with protein kinase C to modulate cell signaling and regulate chemokines and adhesion genes [114,115].

Dietary vitamin E insufficiency is associated with PD in case-control studies [116]; however, when measured, low vitamin E concentrations have been reported in only some [117,118] and not all [101,119,120,121] human PD studies. A recent meta-analysis affirmed this ambivalence of dietary vitamin E status on PD [122]. While vitamin E showed no correlation, other enzymic and low molecular weight antioxidants (urate, catalase, GSH) were found to be reduced in PD together with increased blood oxidation markers [122] thereby maintaining a proposed link between oxidative stress and neurodegeneration. There is some evidence that α-tocopherol supplementation in combination with vitamin C slows disease progression in early-stage PD [123] but this was not recapitulated with α-tocopherol alone [124]. A long-term follow-up of high dose α–tocopherol in early disease patients also failed to demonstrate positive outcomes on mortality [125,126].

Overall, despite positive preclinical studies of neuron loss, inflammation, and oxidation, clinical trials of vitamin E to date have not demonstrated benefit on cognition, neurodegeneration, or mortality in PD. Other vitamin E forms have been tested in neurodegeneration such as tocotrienols (T3s; from palm oil), which have lower bioavailability than α-tocopherol but exert antioxidant and other bioactivities such as mediating cell signaling and are accessible to the brain/CNS. For example, T3s supplementation inhibits dopaminergic neuron loss and improves motor deficits via an estrogen receptor-β mechanism in an MPTP mouse PD model [127] and ameliorates neuronal degradation and reverses locomotor deficits in a 6-OHDA rat model [128]. This suggests that bioactive vitamin E analogs with antioxidative and anti-inflammatory activities may be neuroprotective and should be tested further for efficacy in PD. Tocotrienol-rich vitamin E (Tocovid) possesses potent antioxidant and anti-inflammatory properties and is currently in clinical trials for PD (NCT04491383) [129].

#### 3.2.3. Polyphenols and Flavonoids

Polyphenols are a large group of chemically reactive plant micronutrients credited with the health benefits of fruit/vegetable-rich diets and can be divided into subclasses; phenolic acids, flavonoids, lignans, and stilbenes, of which flavonoids comprise > 60%. Polyphenols demonstrate antioxidant and metal chelating activities in vitro [130] suggesting an ability to moderate neurodegeneration and oxidative processes in the brain. Polyphenols can also augment vascular function by stimulating NO [131] and interact with cell membrane receptors by inhibiting receptor phosphorylation and/or growth factor binding [132]. Dietary polyphenols are associated with reduced risk of dementia [133], cognitive decline [134], and PD [135]. PD locomotor symptoms are significantly delayed by drinking tea containing various flavonoids in a retrospective study [136] and reduction in PD risk was also determined in meta-analyses of tea consumption in case-controlled [137] and prospective cohorts [138] studies. Two additional contemporary prospective studies showed that increased flavonoid consumption, especially the sub-class anthocyanins, are associated with lower mortality in PD [139] and maintenance of cognitive function [140]. Neuroprotection with polyphenol supplementation via anti-inflammatory, anti-apoptotic, and antioxidant activities has been demonstrated in numerous cell and animal PD models [140].

Despite this growing body of literature, the precise mechanism/s whereby polyphenols may be neuroprotective are not well defined. Direct antioxidant scavenging of cellular oxidants may be unlikely given the low bioavailability of polyphenols and limited ability to cross BBB, and moreover, that endogenous antioxidants likely outcompete kinetically. However, at lower concentrations than required for direct antioxidation, polyphenols act on cell-signaling pathways that influence redox status indirectly via antioxidant and cytoprotective gene expression [141]. Indirect antioxidant activities in cells include activation of the Nrf2/antioxidant response element (ARE) pathway that regulates phase 2 antioxidant and detoxification enzyme expression [142,143]. Polyphenols are also low affinity ligands for various cell signal kinases involved in inflammation, mitochondrial respiration, and apoptosis such as AMPK, PI3K/AKT/mTOR pathway, p38 MAPK, and ERK1/2 [144] and kinase dysfunction is genetically linked to PD and thought responsible for hyperphosphorylation of α–synuclein [145].

Antioxidant activities of polyphenols include ROS reduction, maintenance of endogenous antioxidants and mitochondrial function, and a tight interaction between polyphenol antioxidant activity and locomotor function is found in various PD models [146]. For example, green tea polyphenols are neuroprotective in 6-OHDA model via inhibition of ROS, NO, lipid peroxidation, and iNOS [147]. Curcumin increases GSH in the brain and decreases protein oxidation while maintaining complex I activity [148] and SOD1 [149]. Quercetin reduces lipid peroxidation and restores SOD and CAT enzymic activity and GPx/GSH across rodent PD models [150] and increases Na^+^/K^+^ATPase and mitochondrial complex I activity. The latter suggests an ability to repair neurotoxin-induced electron chain defects and intervene in ROS-programmed cell death by alleviating mitochondrial ROS. A natural quercetin derivative, quercetin 3-O-galactoside promotes cell survival by activation of Nrf2 and HO-1 and reducing ROS [151]. MPTP-induced lipid peroxidation and reduced CAT, SOD, and GSH levels are also relieved by naringenin [152]. Epigallocatechin-3-gallate (EGCG), an abundant tea catechol, reduces lipid peroxidation and apoptosis and improves locomotor function in a PD fly model [153]. EGCG also regulates iron export and reduces oxidative stress and protein carbonyl formation in MPTP-treated mice [154] to moderate neuron injury and neuroinflammation.

Anti-inflammation activities of polyphenols are associated with interruption of oxidative injury/cell death cycle via inhibition of α–synuclein aggregation and NFκB-driven cytokine production. For example, green tea polyphenols that modulate NFκB translocation in PD models demonstrate neuroprotection [155]. EGCG increases TH activity and decreases iNOS, TNF–α, IL–6, and nitrite in rodent PD models [140]. Naringenin increases Nrf2, activates ARE genes, and is neuroprotective after 6-OHDA treatment in vitro and in vivo [156]. Downregulation of pro-inflammatory mediators such as TNF–α, IL-β, and iNOS together with reduced α–synuclein aggregates are also observed with naringenin in an MPTP model [152]. Naringin protects dopaminergic neurons by reducing microglial activation and augmenting the mTOR pro-survival pathway [157]. Baicalein suppresses NFkB translocation, and this is accompanied by decreased kinase activity and the promotion of neurogenesis and neurotrophin signaling [158]. Quercetin maintains mitochondrial complex activities, and this aligns with reduced pro-inflammatory biomarkers [121]. Acacetin modulates NO, prostaglandin E2, and TNF-α in vitro and inhibits microglial activation, iNOS, and COX–2, and arrests neuroinflammation and locomotor dysfunction in an MPTP mouse model [159].

Nrf2 and its activation of downstream ARE are therapeutic targets for neuroinflammation and brain redox homeostasis, and several studies support Nfr2 involvement in PD [160,161]. Various polyphenols that promote Nrf2 nuclear translocation are associated with ROS reduction and upregulation of ARE gene expression and GSH with a concomitant reduction in NFκB translocation and downregulation of pro-inflammatory genes [144]. Furthermore, autophagy and mitophagy that support synapse plasticity can be regulated via Nrf2 [162,163]. In addition, there is current interest in neuroprotection by the stress-response protein HO-1, an endogenous antioxidant enzyme induced by Nrf2 that is also therapeutically upregulated by various polyphenols. Plasma HO-1 levels in PD patients are elevated and associated with a reduction in right hippocampal volume [164]. HO-1 is induced by a myriad of natural compounds as well as various polyphenols and flavonoids and this is associated with neuroprotection across PD models [165]. Further, abrogation of neuroprotection is observed with silencing of Nrf2 or inhibition of HO–1 in various cell and animal neurodegeneration studies [156,166].

Individual polyphenols exhibiting multifactorial antioxidant, anti-inflammatory, and anti-apoptotic effects that modulate protein aggregation, ROS, mitochondrial dysfunction, and neuroinflammation to protect neurons and reverse pathological features have been investigated in PD. For example, resveratrol increases antioxidant enzyme capacity and reduces ROS and oxidized lipids, and concomitantly reduces COX–2, PLA2, and TNF–α resulting in improved mitochondrial function and neuroinflammation in 6-OHDA rodent PD models [140]. Resveratrol is thought to interact with cell signaling pathways via AKT/GSK3 and SIRT1/AKT to alleviate mitochondrial dysfunction [167] and enhance autophagic degradation of α-synuclein [168]. Resveratrol has not been tested in PD clinical trials, but two trials in AD (which demonstrates common metabolic dysfunctions and pathogenesis) have shown safe use of resveratrol and penetration of BBB to elicit CNS effects. Thus, decreased CSF MMP–9 degrades extracellular matrix and may protect the CNS [169], causing a slower decline in beta-amyloid (Aβ) peptide, indicating reduced accumulation in the brain [169,170].

Curcumin also increases ARE, such as phase II and antioxidant enzymes and GSH, and reduces lipid oxidation, inflammatory mediators, and iNOS across PD models [140]. In the rat rotenone model, these effects may be mediated via activation of AKT/Nrf2 [171]. Curcumin also moderates glial response, astrocyte activation and reduces NFκB translocation, TNF–α, and IL–1β [172] and inhibits JNK pathways involved in neuronal death [173]. Like resveratrol, there are no trials of curcumin in PD and limited ambivalent trials of its potential in cognitive decline +/− AD [174]. Both curcumin and resveratrol show low absorption and oral bioavailability [175] and this may affect therapeutic potential particularly in neurodegeneration. However, one study of low-dose lipidated-curcumin complex in a healthy cohort reported decreases in plasma levels of triglyerides, Aβ, sICAM, and ALT with a concomitant increase in the salivary antioxidant capacity and plasma CAT, NO, and MPO in the absence of changes in the circulating inflammatory marker CRP. The latter was attributed to a strengthened immune response and overall, the intervention indicated a positive effect of supplementation on health promotion [176]. Multifactorial and off-target effects of polyphenols may be a concern, and the metabolism of these natural products can also further limit their bioactivity and bioavailability. Further investigations that show significant polyphenol delivery to plasma/CSF with functional effects in CNS will be important.

### 3.3. Bioactive Natural Herbal Extracts 

The use of natural herbs in the treatment of neurodegenerative diseases has been widely recognized by Traditional Chinese Medicine and is often associated with few (or low incidence of) adverse effects in humans. The presence of polyphenols and related phytochemicals with biologically active properties in these herbal preparations is increasingly a focus in contemporary research, especially in the field of developing new complementary therapies [177] that may provide therapeutic benefits. For example, Kurarinone, a major constituent of the *Sophorae Flavescentis Radix*, (referred to as Kushen in China), is commonly used to treat diarrhea and inflammation-related disorders [178]. Additionally, 4′,Polydatin (3,4′,5-trihydroxystilbene-3-β-D glucoside), a natural resveratrol glucoside extracted from *Polygonum cuspidatum,* exhibits pharmacological effects against degenerative motor function by protecting dopaminergic neurons in the rat substantia nigra [178]; here polydatin decreased microglial activation through stimulating the AKT/GSK3β-Nrf2 signaling axis to ameliorate lipopolysaccharide-mediated experimental PD [179] indicative of an anti-inflammatory action for polydatin.

Current pharmacological research indicates that Astragaloside IV (AS-IV), an antioxidant derivative of *Astragalus membraneaceus Bunge*, can both relieve inflammation and improve behavioral disorders in an MPTP-induced mouse model of PD [180]. Yet, other studies in cell and animal models of PD have reported that herbal medicines can diminish neuronal inflammation, oxidative stress, and/or the level of apoptosis when taken alone [181,182] or when co-administered with conventional medicines [183]. Furthermore, a substantial body of evidence indicates that herbal medicines can attenuate neuroinflammation in PD through diminishing iNOS-dependent nitric oxide release (thereby down-regulating the formation of damaging ONOO-), COX–2 activity in glial cells, unregulated ROS generation, and decreasing the expression of inflammatory cytokines such as TNF-α and IL–6 [178,184,185]. Therefore, there is increasing interest in identifying natural herbal treatments that represent a holistic approach to limit the impact of PD in humans.

Studies have described herbal tonics that prevent dopaminergic neurodegeneration through the suppression of microglia activation [179] and inhibition of astrocyte senescence [179]. Importantly, several studies have indicated that these protective activities involve a common suppression of NFκB signaling pathways. In LPS-induced mice, the administration of 10 mg/kg of Icariside II, an active flavonoid extracted from *Epimedium*, diminishes inflammation by increasing IκB–α protein accumulation in the hippocampal tissues, ultimately ameliorating NFκB transcriptional activation as judged by monitoring tissue levels of NFκB-p65 phosphorylation [186]. Similarly, the isoflavone, Tectorigenin, isolated from various medicinal plants, such as *Pueraria thunbergiana Benth*, *Belamcanda chinensis*, and *Iris unguicularis* can suppress the NFκB/ERK/JNK molecular signaling axis, which leads to a down-regulation of inflammatory mediators such as TNF-α, iNOS, COX–2, and IL–6 [185] and improves cellular viability. Also, the *Artemisiae Iwayomogii Herba*, locally known as ‘Haninjin’, is reported to inhibit the formation of the multiprotein complex NLRP3/ASC/Caspase 1 inflammasome in microglia by directly suppressing expression of NLRP3, a protein that is also regulated by NFκB transcriptional activation and ensuing MAPK signaling [187].

Alternate mechanisms of action include neuroprotection offered by traditional medicinal plants through enhancing endogenous antioxidant enzymes systems such as SKN–1/Nrf2 activation via the MAPK pathway in a *Caenorhabditis elegans* model of PD [188]. More recent studies have implicated a key role of soluble epoxide hydrolase (sEH) enzyme and its biological substrates such as eicosatrienoic (EETs) and epoxydocosapentaenoic (EDPs) acids in the regulation of neuroinflammation [189]. Thus, pharmacological inhibition of sEH (or through sEH gene deletion) protects the mouse brain from MPTP-mediated neurotoxicity; notably, a positive correlation between sEH expression and the extent of α-synuclein phosphorylation was identified in the striatum tissue in this experimental model of PD. In strong support for this pathway in humans, patients with Lewy body-dementia and other chronic psychiatric diseases showed elevated sEH protein in post-mortem brains [190,191], further implicating sEH in the pathogenesis of neuroinflammation that is characteristic of PD. In the same MPTP-induced PD mice model, the traditional Chinese herbal medicine *Sophora flavescens* alleviated dopaminergic neurotoxicity including loss of neurotransmitters and expression of TH in the substantia nigra and striatum and this resulted in an improved gait. Interestingly, kurarinone, the active constituent of *Sophora*
*flavescens*, showed an inhibitory effect against sEH by interacting with amino acid residues, stimulating an increase in plasma levels of EETs and a corresponding decrease in its oxidation product(s) such as dihydroxyeicosatrienoic acids [178]. 

Despite the success of numerous in vivo and in vitro studies, clinical trials are limited and largely incipient. A summary of available recent trials is collected in Table 1. In 2020, Chen, S.Y. et al. [192], studied in seventy-two patients with PD the clinical efficacy of Congrongjing granules, mainly composed of *Herba cistanches* and *Polygonatum sibiricum.* Administration for twelve consecutive weeks improved quality of life, alleviated symptoms and reduced the dosage of Levodopa in patients with PD without obvious serious side effects or obvious worsening of clinical symptoms. Studies have also shown that cell metabolism-related processes and ribosome-related pathways are central to translation to clinical efficacy. In a total of one hundred and seven patients with PD and sleep disorders, treatment with Huatan Jieyu granules for four weeks significantly improved both the individual’s sleep time and the stages of sleep period, with persistent benefit reported three and six months after cessation of the treatment [193]. A multi-center, double-blind, placebo controlled RCT with approximately two hundred and forty participants is currently underway to examine the efficacy and safety of SQJZ herbal mixtures on non-motor symptoms (NMS) of PD when taken two times daily over a period of twelve weeks [194]. This study is in Phase 2 and the future findings from this research may provide for the first-time clinical evidence on the use of the SQJZ formula in NMS in PD patients (https://clinicaltrials.gov, accessed on 7 April 2022). 

Overall, the potential therapeutic effects of extracts and bioactive components from medicinal herbs for the treatment of PD may represent a novel pharmacological holistic approach to decrease oxidative and inflammatory pathways in the PD brain, which may translate to clinical benefit. Complications in the use of natural herbal products include (i) the (seasonal) variation of active constituents in harvested materials, (ii) metabolism of precursor components that may vary from individual to individual, and (iii) the complexity of the extract/tonic, which contains many different potentially active components that may act in concert (synergistically) or individually to achieve therapeutic advantage. Thus, further large-scale RCTs are required to rigorously demonstrate an unambiguous benefit for these traditional herbal medicines in the setting of PD.

### 3.4. Synthetic Antioxidants

Although potential disease-modifying roles for endogenous and dietary antioxidants have not been realized in human PD (as discussed above) there is still ongoing interest in synthetic antioxidants as therapeutics, especially compounds that target the mitochondria to quench ROS. For example, an oral derivative of mitochondrial-targeted CoQ_10_, mitoquinone (MitoQ_10_), protects against oxidative stress in a wide range of disease models and exhibits positive effects on mitochondrial function and stability in both cell and animal models of PD together with neuroprotection and reversal of behavioral deficits [69]. Although promising, MitoQ_10_ has not shown benefit to date as it may be unlikely to affect clinical remission of PD [69] and, therefore, requires further trials to establish therapeutic potential. Other mitochondrial-targeted synthetic antioxidants have been developed based on carbon-centered radical spin traps (MitoPBN), ebselen (Mitoperoxidase; that can chemically reduce and recycle oxidized glutathione disulfide), and SOD dismutation (MitoSOD) [69]. Of these, the SOD mimetic M40403 shows neuroprotective potential and protects against cellular oxidative damage and compensates for the loss of cytosolic and mitochondrial SOD, improves locomotor behavior [198], and partially ameliorates PINK–1/Parkin phenotypes [199] in an in vivo model of PD.

A further group of synthetic antioxidants with bioactivity in preserving mitochondrial function and redox homeostasis are cyclic nitroxides which are discussed in detail in the next section. Cyclic nitroxides have been demonstrated to exert significant neuroprotective activities and therefore may be beneficial in PD. For example, cyclic nitroxides mitigate loss of antioxidant enzyme and mitochondrial function and improve motor impairments in neurodegenerative disease [200] and minimize apoptosis and improve motoneuron and synapse survival after injury [201]. Cyclic nitroxides also improve brain iron homeostasis and neuromuscular impairment [202] and mitochondrial function and neurobehavioral outcomes after anoxia [203]. Recently, a cyclic nitroxide was shown to modulate inflammation and axonal damage and delay onset and reduce disease burden in an animal model of multiple sclerosis [204].

#### Cyclic Nitroxides

Cyclic nitroxides are stable free radicals that can act as antioxidants and anti-inflammatory agents by modifying the redox state of tissues and inhibiting post-translational oxidative modification of proteins. This class of stable free radicals undergoes reversible one- and two-electron transfer reactions between the nitroxide radical, the reduced hydroxylamine, and the oxidized ammonium cation form when reacting with a range of ROS [205]. This is facilitated by biological substrates and ‘boat and chair’ conformational changes, making this class of drug an ideal ROS scavenger and inhibitor of ROS-mediated damage [206].

Nitroxides are well placed to exert their antioxidative effects, since mitochondria, due to their low partial pressure of oxygen and high ROS production, are the major site of reduction of the nitroxide radical to the hydroxylamine [205]. By shuttling between the nitroxide radical/oxoammonium cation redox pair, nitroxides act as superoxide dismutase (SOD) mimetics that convert superoxide anion radicals into hydrogen peroxide which can then be scavenged by peroxidases in a cyclic process that deactivates the oxidizing potential for these ROS [206]. Although the catalytic rate of this SOD-mimetic reaction is less than that of the superoxide dismutase enzyme itself, nitroxides have a low molecular weight and accumulate intracellularly such that their relative effectiveness as a SOD mimetic is comparable due to the differences in concentration between the drug and the actual SOD enzyme [206].

Furthermore, nitroxides enhance the catalase-like activity of heme proteins as they can detoxify the hypervalent heme protein, formerly a ferryl iron-IV-oxo species, which is a product of the reaction between hydrogen peroxide and a native peroxidase enzyme [206]. Similarly, by substituting themselves into the redox reaction and accepting the electron from the reduced metal complex instead, nitroxides prevent the free ferrous iron-catalyzed conversion of superoxide anion radicals to hydrogen peroxide in the Fenton reaction [206]. However, dose is an important consideration, as high concentrations of TEMPOL have been shown to affect the assimilation of ferrous iron into ferritin, increasing the concentration of damaging free ferrous iron and potentially impacting important iron transport pathways [207].

Lipid peroxidation is a free radical or redox-active transition metal complex activated chain reaction that results in membrane damage, oxidative modification of critical molecules, and cellular injury [206]. Nitroxides inhibit both the initiation and propagation phase of the chain propagation reactions by interacting with the free radicals involved, thereby inhibiting the chain-carrying species and limiting damage to lipid bilayers and cell membranes [206]. Therefore, nitroxide-mediated inhibition of lipid peroxidation may represent one mechanism of neuroprotective activity in the setting of PD.

The anti-inflammatory mechanisms of action of nitroxides include inhibition of the myeloperoxidase (MPO)-hydrogen peroxide-chloride system of activated microglia, as well as the oxidation of the highly toxic ONOO- which is produced by the reaction between microglia released superoxide radical anion and nitric oxide radicals [208,209]. In addition, nitroxides act as general scavengers of ROS and thus reduce the initiation of the sphingomyelin- or ceramide-dependent signaling pathway that results in NFkB translocation and apoptosis induction. Indeed, this mechanism of inhibiting apoptotic cell death may represent a significant pathway that explains the neuroprotective activity of cyclic nitroxide drugs.

In direct support of the potential therapeutic action of nitroxides, we have shown that 4-hydroxy-2,2,6,6-tetramethylpiperidin-1-oxyl (TEMPOL) ameliorates 6–OHDA induced neurotoxicity in a differentiated SH-SY5Y cell model and that this mechanism of neuroprotection is mediated by attenuation of oxidative stress and mitochondrial dysfunction [210]. However, contrary to the hypothesized anti-inflammatory properties of TEMPOL, this cyclic nitroxide did not affect markers of inflammation such as IFNγ and IL–6 [210]. Similarly, TEMPOL has been shown to inhibit 6–OHDA mediated apoptosis and NFkB activation in MN9D dopaminergic mesencephalic cells and in a murine model, reduce striatal cell and catecholamine metabolite loss as well as the severity of parkinsonian symptoms, and autonomic dysfunction [211]. Furthermore, consistent with the antioxidant mechanism of nitroxides, TEMPOL reduced apoptosis of TH-positive cells located in the substantia nigra in mice deficient in apoptosis-inducing factor, a mitochondrial protein vital to the functioning of complex I [212].

TEMPOL is effective after oral administration in affecting the CNS as it is an ampholyte with a high capacity to permeate cell membranes and readily crosses the BBB, allowing the drug to accumulate in brain tissue [204]. However, a growing area of research is in the use of nanoparticles, especially in neurodegenerative disease, as nanoparticle-based delivery systems for nitroxides have better BBB penetration and more favorable pharmacokinetics [213,214]. In undifferentiated SH-SY5Y cells, the cyclic nitroxides, TEMPO and 4-amino-TEMPO, as well as nitroxide-containing redox nanoparticles (NRNPs) protected against mitochondrial dysfunction, ROS, and superoxide production and reduced cell viability induced by 6–OHDA [215]. Notably, the NRNPs employed in this study were found to better model BBB penetration and antioxidant activity than the free nitroxides investigated [216]. Furthermore, other strategies to enhance the therapeutic index of cyclic nitroxides exist such as polynitroxylated albumin (PNA), which facilitates the recycling of inactivated antioxidants to their redox-active form [217]. The antioxidant complex, TEMPOL/PNA was less toxic than TEMPOL alone and more effective in preventing 6–OHDA mediated depression of mice activity as well as apoptosis and NFkB activation in undifferentiated PC12 cells [215]. Nitroxides are a promising potential therapeutic approach to PD with demonstrated neuroprotective effect through a mechanism of reducing oxidative stress and mitochondrial dysfunction, although further establishment of these conclusions, especially in animal models, is necessary before progression to clinical trials.

## 4. Conclusions

Oxidative stress is a prominent pathophysiological mechanism inherent in the development of PD as ROS and the dysfunction of antioxidant systems both directly result in dopaminergic neuron loss and lead to the downstream disruption of numerous cellular systems, including mitochondria, protein formation, and regulation of inflammation (refer to Figure 1). Accordingly, antioxidants were hypothesized as promising therapeutic approaches to modifying the underlying pathogenesis of PD. However, despite encouraging results in cell and animal models, various antioxidants, such as CoQ_10_, urate, glutathione, and vitamin C have not demonstrated similar efficacy as this avenue of research progressed to clinical trials. This discrepancy is likely reflective of the complex nature of the human disease, especially where early and asymptomatic PD occurs in humans, which is poorly represented in currently available cell and animal models. 

Despite these limitations, the testing of new natural products continues, and potential therapies based on biologically active extracts and traditional herbal approaches have gathered momentum in the search for therapeutic targets against neurodegenerative disorders such as PD. Interestingly, the recent development of novel cellular models such as human-brain based organoids [217,218] and patient-derived pluripotent stem cells [219] may provide additional experimental models that have greater translation capacity for therapeutic development [220] than the models described above. For example, microglia integration with human-brain based organoids is demonstrated to enhance neuronal maturation and functionality in these three-dimensional structures [221] and may provide a superior experimental model to test drug/antioxidant efficacy, although to date no data has been made available for evaluation. Overall, further clinical trials are necessary to further elucidate the therapeutic potential of antioxidants. The development of novel treatment strategies, such as nanoparticle delivery systems and the use of individually tailored, omics-based approaches, as well as targeting early disease intervention with adequate models, indicates that the use of antioxidants for the treatment of PD remains a field of research.

## Figures and Tables

**Figure 1 ijms-23-06923-f001:**
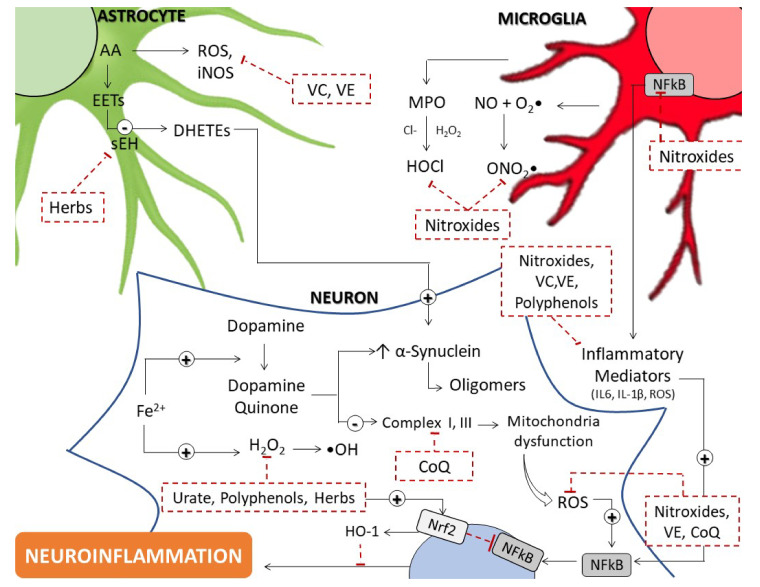
Schematic figure showing the role of oxidative stress, unregulated generation of reactive oxygen species (ROS) and neuroinflammation in the pathogenesis of Parkinson’s Disease. This figure summarizes various molecular pathways identified and implicated in the unregulated generation of ROS, ensuing oxidative stress and inflammatory pathways in neurodegeneration and PD. How natural and synthetic agents target these pathways has led to varying outcomes in experimental models of PD, yet few human trials have shown benefit and in the case of natural herbal therapies, rigorous extensive trials have not yet been conducted. AA: Arachidonic acid, CoQ: Coenzyme Q, DHETEs dihydroxyeicosatrienoic acids, EET: Epoxyeicosatrienoic acid, HO–1: Heme oxygenase–1, iNOS: inducible Nitric Oxide Synthase, MPO: Myeloperoxidase, NFkB: Transcription factor nuclear factor kappa B, Nrf2: nuclear factor erythroid 2–related factor, ROS: Reactive oxygen species, sEH: Soluble epoxide hydrolase, VC: Vitamin C, VE: Vitamin E.

**Table 1 ijms-23-06923-t001:** Summary of recent clinical trials that are published or currently underway (https://clinicaltrials.gov, accessed on 3 June 2022).

N Patients	Recruitment Selection Strategy	Study Design	Primary Outcomes	Secondary Effects	Status
CoQ10					
20	-Stable medication with carbidopa/LD, Selegiline and anticholinergic medication. -Symptoms of PD from 1–17 years. -54 to 70 years; All sexes	**4 groups:** CoQ_10_ provided at doses of 300, 600 or 1200 mg/day plus vitamin E (300 UI) and a placebo.	-↑ of plasma CoQ_10_ level -Normalize Complex I activity mitochondrial -↓ decrease in the original UPDRS score	Nonsignificant secondary effects	Phase 2 **NCT00004731** Ref [45]
600	-Without PD medication before 60 days. -Presence of all cardinal signs within 5 years. -Hoehn and Yahr stage of 2.5 or less -30 years and older; All sexes	**3 groups:** CoQ10 at doses of 1200 and 2400 mg/day plus vitamin E (1200 UI) and placebo	No change in MDS-UPDRS score, Hoehn and Yahr score, PDQOL and SE-ADL scales.	-Back pain, -Constipation, -Anxiety, -Headache	Phase 3 **NCT00740714** Ref [47]
72	-Stable PD medication for at least 4 weeks -18 years and older -All sexes	**4 groups:** Stratified by their genetic mitochondrial risk burden and randomized 1:1. CoQ10 at dose of 156 mg/day [QuinoMit Q10^®^ Fluid] and placebo	Not reported	Not reported	Phase 2 **DRKS00015880** Ref [55]
**Urate/Inosine**					
298	-Without PD medication -Serum urate ≤ 5.7 mg/dL at 1st visit -Presence at least 2 of the cardinal signs -Hoehn and Yahr stage of 2.5 or less -30 years and older; All sexes	**2 groups:** Inosine at dose of 500 mg, 1 to 6 capsules per day to achieve serum urate at 7.1 to 8.0 mg/dL and placebo for 24 months	-No change in MDS-UPDRS score and DAT binding -↑ of serum urate level.	-Kidney stones	Phase 3 NCT00833690 Ref [73]
75	-Without PD medication -Serum urate ≤ 5.8 mg/dL at 1st visit -Presence at least 2 of the cardinal signs -30 years and older; All sexes	**3 groups:** Inosine at dose of 500 mg, 1 to 6 capsules per day to achieve serum urate at 6.1–7.0 mg/dL (mild), 7.1–8.0 mg/dL (moderate) and placebo for 2 years	-↓ significant decrease in the MDS-UPDRS score in moderate group in woman. -↑ plasma antioxidant capacity in woman. -↑ serum urate level and CSF in women	-Kidney stones	Phase 2 NCT02642393 Ref [195]
**Glutathione/N-acetyl-cysteine**					
42	-Stable PD medication for at least 1 month -Hoehn and Yahr score of 1–2 inclusive -30 to 80 years -All sexes	**2 groups:** IV NAC at dose of 50 mg in 200 mL at frequency over one hour 1 × per week for 90 days plus Oral NAC at dose of 1200 mg/day and standard of care treatment (control group)	-↓ significant decrease in the MDS-UPDRS score -↑ DAT binding in caudate and putamen	Not reported	Not applicable **NCT02445651** Ref [86]
8	-Stable PD medication -18 years and older -All sexes	**2 groups:** NAC at dose 6000 mg/day for 4 weeks and healthy controls	-↑ significant increase in catalase and GSH/GSSG peripheric -No change in 4–HNE and MDA. -No change in brain levels of GSH	-Gastrointestinal disorder -Nervous system disorders	Phase 2 **NCT02212678** https://clinicaltrials.gov accessed on 3 June 2022
35	-Stable medication with monoamine oxidase inhibitor. -Diagnosis within the past 5 years -18 years and older; All sexes	**2 groups:** NAC at dose 2 g/day for 3 days and healthy controls	-↑ Cys–DA/DOPAC ratio	Not reported	Phase 1 **NCT03104725** Ref [196]
47	-Without PD medication -50 to 75 years -All sexes	**3 groups:** NAC at dose of 1800 mg/day, 3600 mg daily and placebo for 30 days	Not reported	Not reported	Phase 2 **NCT01470027** https://clinicaltrials.gov accessed on 3 June 2022
**Vitamin C**					
67	-Stable medication with LD -64 years and older -All sexes	**1 group:** First time: 100 mg LD + 10 mg carbidopa and Second time (1 week later) 100 mg LD + 10 mg carbidopa + 200 mg AsA.	-↑ LD pharmacokinetics	Not reported	Not applicable Ref [98]
**Vitamin E (Tocotrienols)**					
100	-Stable PD medication -Diagnosis within the past 1 year -PD of more than 1 year duration from diagnosis. -Hoehn and Yahr score ≥ 2 -40 to 90 years -All sexes	**2 groups:** Tocotrienols (400 mg/day) and placebo for 12 months	Not reported	Not reported	Phase 2 **NCT04491383** https://clinicaltrials.gov accessed on 3 June 2022
**Polyphenols**					
38	-Stable PD medication for at least 1 month. -Hoehn and Yahr score of 1–3 -30 years and older -All sexes	**3 groups:** EGCG at dose of 800 mg, dose of 1200 mg and placebo (mannitol) group for 4 weeks. Then one capsule twice daily for 4 weeks, and then one capsule three times daily for 40 weeks	-No change in UMSARS score	-Hepatotoxicity	Phase 3 **NCT00461942** https://clinicaltrials.gov accessed on 3 June 2022
**Herbal Medicine**					
30	-Stable PD medication -Initiation of PD symptoms in recent 6 years -Hoehn and Yahr score of ≤3 -30 to 80 years	**2 groups:** Oral Licorice at dose 10 cc/day and placebo (syrup) for 6 months	-↓ significant decrease in the MDS-UPDRS score -↓ daily activities score -↓ motor test score -No change in tremor, rigidity score and YAHR score.	-Nausea -Diarrhea -Urticaria	Not applicable Ref [197]
72	-Stable PD medication -Hoehn-Yahr score ≤4 -50 to 80 years; All sexes	**2 groups:** CSGs (*Herbal cistanches* 6 g, *Polygonatum sibiricum* 12 g, *Salvia miltiorrhiza Bge.* 15 g, *Radix Paeoniae rubra* 12 g, and *Cortex Moutan* 10 g), 2 capsules per day and placebo (*Radix Bupleuri*) for 12 weeks.	-↓ significative in the MDS-UPDRS sub-II score, -↓ significative in PDQ-39 score, -Significative improvement in CM syndrome score.	-Internal heat -Thirsty -Dryness heat.	**ChiCTR-IOR-16008394** Ref [192]
107	-Basic treatment with prednisone. -Sleep disorder	**2 groups:** Huatan Jieyu granule and placebo for 4 weeks.	-↑ significative of total sleep time -↑ significative of NREM 2-3 -↑ significative of REM sleep -↓ significative in NREM1	Not reported	Not applicable Ref [193]
240	-Stable medication with carbidopa/LD, anticholinergics, MAO inhibitors, or amantadine for at least 28 days -Hoehn and Yahr stage score of ≤4 -NMSS score ≥40 -18 to 80 years; All sexes	**2 groups:** SQJZ herbal mixtures at dose of 29.375 g, 2 times per day and placebo for 12 weeks	Not reported	Not reported	Phase 2 NCT02616120 https://clinicaltrials.gov accessed on 3 June 2022

AsA: Ascorbic acid; CoQ_10_: Coenzyme Q_10_; CSF: cerebrospinal fluid; CSG: Congrong Shujing Granules; DAT: Dopamine transporter; DOPAC: 3,4- dihydroxyphenylacetic acid Cys 5-S-cysteinyl-dopamine; EGCG: Epigallocatechin gallate; IV: Intravenous; LD: Levodopa; MAO: Monoamine oxidase; MDS: Movement Disorder Society; NREM: Non rapid eye movement; NMSS: Non-Motor Symptoms Scale; NAC: N-acetyl-cysteine; PDQOL: Parkinson Disease Quality of Life; REM: Rapid eye movement; SE-ADL: Schwab and England -activities of daily living; UPDRS: original Unified Parkinson’s Disease Rating Scale; MDS-UPDRS: Muscle Disorder Society updated Unified Parkinson’s Disease Rating Scale; arrows correspond to: ↑: Increase; ↓: Decrease.

## Data Availability

Not applicable.

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
