# Peer review of "Evidence for Oxidative Pathways in the Pathogenesis of PD: Are Antioxidants Candidate Drugs to Ameliorate Disease Progression?"

_ijms, 2022, doi:10.3390/ijms23136923_

Round 1

Reviewer 1 Report

In their very informative and nicely written review, Leathem and colleagues describe current evidence on the involvement of oxidative stress in pathogenesis of Parkinson's disease and review preclinical and clinical data on the application of selected antioxidants for the treatment of neurodegeneration. The manuscript is very interesting and certainly will be appreciated by many readers in the field. I especially like the very informative figure illustrating major oxidative stress pathways affecting dopaminergic neurons and the strategies to target those. However, I still  have several suggestions for potentially improving the manuscript further.

The authors describe several past and ongoing clinical trials addressing efficacy of various antioxidant compounds in PD. It would be very informative to include the table summarizing these clinical trials with outcomes (if they are known) and corresponding references. 

Also, the authors rightfully mention the problem of poor translational value of current preclinical animal models, however, they do not highlight any recent attempts to solve this problem. It would be very interesting to know the author's opinion on utilizing human iPS cell-derived neurons and brain organoids in preclinical studies. Have antioxidants been tested in such models and what are the results? Would human midbrain organoids (which can now include also microglia cells; see, for example PMID 35262217) potentially provide a model with better translational value? 

Several sentences in the text appear to be broken, please, proofread carefully. 

Author Response

Dear Editors IJMS

Please find here a resubmission for the manuscript IJMS_1732273 as an overview of contemporary therapeutic approaches to Parkinson's disease following our recent publication in Antioxidants.  We have carefully assessed reviewer comments and adjusted the text to comply with all suggestions as outlined in the accompanying point-by-point response document.

This revised review manuscript was composed, edited, and reviewed by all the named authors and all have agreed to the submission to IJMS.

No part of this review article has been published elsewhere or is being considered by another journal.  The figure included as a schematic is original and outlines the various approaches discussed in this contemporary review of the literature on Parkinson's Disease.  We also include 1 Table within the body of text.

We appreciate the time and effort required to review the submission and look forward to receiving Editorial and reviewer comments in due course.

Kind regards

Paul

Paul Witting

Professor; Redox Biology

School of Medical Sciences

Faculty of Medicine and Health

The University of Sydney

Reviewer 2 Report

This is a nice paper tackling an important issue of the possible use of antioxidants to ameliorate progression of PB.It seems that the paper is thorough and well written.

However, I'd like to see a Table of the studies (current and past) that trialed drugs with antioxidative action to test the their neuroprotective action. The authros can have a look at this paper as an example.

Vijiaratnam N, Simuni T, Bandmann O, Morris HR, Foltynie T. Progress towards therapies for disease modification in Parkinson's disease. Lancet Neurol. 2021 Jul;20(7):559-572. doi: 10.1016/S1474-4422(21)00061-2. PMID: 34146514.

Author Response

(The authors gave the same response as above.)

Author Response

(The authors gave the same response as above.)
